# Antimicrobial Susceptibility of Fresh Produce-Associated Enterobacteriaceae and Enterococci in Oman

**DOI:** 10.3390/foods11193085

**Published:** 2022-10-05

**Authors:** Zahra S. Al-Kharousi, Nejib Guizani, Abdullah M. Al-Sadi, Ismail M. Al-Bulushi

**Affiliations:** 1Department of Food Science and Nutrition, College of Agricultural and Marine Sciences, Sultan Qaboos University, P.O. Box 34, Al-Khod 123, Oman; 2Department of Plant Sciences, College of Agricultural and Marine Sciences, Sultan Qaboos University, P.O. Box 34, Al-Khod 123, Oman

**Keywords:** antimicrobial resistance, chlorhexidine, *Enterobacteriaceae*, enterococci, food safety, fruits, vegetables

## Abstract

Fresh produce bacteria may have phenotypic and/or genotypic antimicrobial resistance traits that may lead to various consequences on the environment and human health. This study evaluated the susceptibility of fresh produce bacteria (banana, cabbage, capsicum, carrots, cucumber, dates, lettuce, mango, papaya, pomegranate, radish, tomato and watermelon) to chlorhexidine and the antibiotic resistance of enterococci. Eighty-eight Enterobacteriaceae bacteria and 31 enterococci were screened for their susceptibility to chlorhexidine using the broth microdilution method. Susceptibility of enterococci to various antibiotics was determined using agar dilution, colorimetric, and Kirby-Bauer disc diffusion methods. Enterococci were more susceptible to chlorhexidine than Enterobacteriaceae indicated by chlorhexidine minimum inhibitory concentration (MIC) of 1 to 8 µg/mL for the former and 1 to 64 µg/mL for the latter. The *IntI 1*, *qacEΔ1*, *qacE* and *qacG* genes were distributed weakly in three, two, two, and three Enterobacteriaceae isolates, respectively. Enterococci had resistance to chloramphenicol (3%), tetracycline (19%), erythromycin (68%), ciprofloxacin (55%), and vancomycin (10%) while 19% of them were multi-drug resistant. In conclusion, this research detected a low to moderate level of antibiotic resistance in enterococci. Some Enterobacteriaceae bacteria had reduced chlorhexidine MICs that were not 10x less than the recommended concentration (100–200 µg/mL) in food production areas which might challenge the success of the disinfection processes or have clinical implications if the involved bacteria are pathogens. The prevalence of antimicrobial-resistant bacteria in fresh produce should be monitored in the future.

## 1. Introduction

Biocides are agents used to kill, inhibit or control the growth of microorganisms. Antiseptics are biocides that inhibit or kill microorganisms associated with living tissue while disinfectants are biocides that are mostly used on inanimate things or surfaces for the same purpose. Biocides are important for any infection control system [1], and although they are widely used;, the resistance of bacteria to them is less documented in the literature than antimicrobials [2]. Recently, there has been an increasing trend in using disinfectants and antiseptics for decolonization of pathogens from inanimate surfaces and the human body [3]. This might lead to the emergence of resistant bacteria to the decolonizing agents [4,5]. In the food industry, disinfectants are extensively used for hygienic purposes to achieve safer foods with longer shelf life. Bacteria that survive the disinfection process can cause food spoilage or affect a food process such as fermentation, and thus may have an economic impact. They may also cause disease if they are pathogens [6] and influence the morbidity and mortality rates as they challenge the prevention of disease transmission [7]. For instance, *Listeria monocytogenes* and *Staphylococcus* spp. isolated from food and food processing plants showed reduced susceptibility levels to disinfectants [8]. Oxidizing agents, phenolic compounds, quaternary ammonium compounds (QACs), and biguanide are the most common classes of biocides used in the food industry [9].

Disinfectants and other non-antibiotic agents have multi-target sites that are mostly located inside microbial cells [10,11]. Resistance to disinfectants can occur through intrinsic or acquired mechanisms [11]. The intrinsic mechanism is a natural, chromosomally encoded process [1]; examples include cellular barriers and efflux pumps that pump antimicrobials outside the bacterial cells [12]. Acquired resistance occurs when there is a change in the genetic makeup of the cell introduced by mutation or horizontal gene transfer of genetic determinants that code resistance to disinfectants [13]. Selective pressure caused by the widespread use of QACs may cause the dissemination of *qac* resistance genes among bacteria in different niches [14]. The *qac* genes encode membrane-embedded proteins which are efflux proteins that resist the effects of QACs [6].

Chlorhexidine (1,1′-hexamethylenebis [5-(4-chlorophenyl)-biguanide]) has become one of the most used antimicrobials nowadays [4]. It is used in food production areas because of its low toxicity and non-corrosive action [6]. Resistance to chlorhexidine can occur through alteration in membrane permeability or by the active efflux pumps [15,16] whose genes are commonly located on mobile genetic elements [15]. It has been shown that decolonization with chlorhexidine in the intensive care unit (ICU) has led to the selection of methicillin-resistant *Staphylococcus aureus* (MRSA) with reduced susceptibility to chlorhexidine [3].

Antibiotics are chemical compounds produced originally by microorganisms and used to prevent or treat infections in humans and animals [17,18]. Antibiotic resistance describes the inherited ability of a microorganism to grow at high levels of an antibiotic, irrespective of the duration of treatment. Resistance of pathogens to antibiotics is costly and can lead to treatment failure [10]. Antibiotics are largely used in food production [19], and the food chain is an important route for transferring antibiotic-resistant bacteria or their genes to humans [20]. Co-resistance or cross-resistance might occur between disinfectants and antibiotics leading to isolates that are resistant to both of them [8]. A ‘post-antibiotic era’ might happen when previously treatable common infections result in life-threatening diseases [21,22]. Therefore, monitoring antibiotic and disinfectant resistance in different environmental and clinical settings is important to manage this issue.

Many outbreaks linked to the consumption of fresh produce are due to bacteria belonging to Enterobacteriaceae and many members of this group have acquired resistance to most antibiotics [13,23]. Enterococci are widely distributed in nature and can be found in the gastrointestinal tract of animals and humans and in foods that originated from animals or plants. They have been involved in food intoxication, nosocomial infections, and spreading antibiotic resistance through the food chain [24].

The role of fresh fruits and vegetables in harboring and disseminating antibiotic-resistant Enterobacteriaceae was discussed in our previous paper [25]. The present study aims to determine the susceptibility of fresh produce-associated Enterobacteriaceae and enterococci to various antimicrobials including those that have clinical significance. The findings from this study can widen our understanding of the antibiotic resistance status or the presence of reduced susceptibility to chlorhexidine in fresh produce bacteria which can help control them in the future.

## 2. Materials and Methods

### 2.1. Sample Collection and Bacterial Isolation and Identification

Samples were collected and analyzed microbiologically, as was previously described [25]. Briefly, thirteen types of fresh fruits and vegetables were obtained from the local markets in Oman and included 39 local and 66 imported samples. The samples included watermelon, tomato, radish, pomegranate, papaya, mango, lettuce, dates, cucumber, carrot, capsicum, cabbage, and banana. Enterobacteriaceae bacteria were isolated on Violet Red Bile Glucose (VRBG) Agar, *Escherichia coli* on Tryptone Bile X-glucuronide (TBX) medium, and enterococci on Slanetz Agar (SA). At least three bacterial isolates showing the typical colonial morphology were selected to represent each positive sample and were purified by subculturing onto Tryptic Soy Agar (TSA). The bacteria were initially identified by VITEK2- compact 15 (bioMérieux, Marcy L’Étoile, France) and then using Polymerase Chain Reaction (PCR) targeting bacterial 16S rRNA gene as was described in our previous articles [25,26]. Nonduplicate bacterial isolates that originated from different samples were included in the susceptibility tests (88 Enterobacteriaceae (38 local, 50 imported) and 31 enterococci (12 local, 19 imported). The accession numbers of Enterobacteriaceae were from KR265345 to KR265359 and from KR265363 to KR265470, while those of enterococci were from KR265363 to KR265393 as were assigned by the GenBank.

### 2.2. Susceptibility of Bacteria to Chlorhexidine

There is no standard method for the determination of chlorhexidine resistance in the literature [15]. In this study, the broth microdilution method [2] was used to determine the MIC. Bacteria were grown on TSA for 18–24 h. Bacterial suspensions were made in Maximum Recovery Diluent (MRD) and the density was adjusted to 0.5 McFarland standard using the DensiCHEK (BioMérieux, France), then diluted by transferring 100 μL of the homogenous bacterial suspension to 9.9 mL Muller Hinton Broth (MHB). A stock solution of 5000 μg/mL (0.5%) of chlorhexidine (chlorhexidine digluconate solution 20% (*w*/*v*) in H_2_O, Sigma-Aldrich, Madrid, Spain) was prepared in sterile distilled water and then serial 10 twofold dilutions of chlorhexidine were prepared in MHB (0.25 to 128 μg/mL or 0.025 × 10^−3^–12.8 × 10^−3^%) and 50 μL from each dilution was delivered to a well in a 96-well microtiter plate (wells 1–10). Fifty microliters of the test bacterial suspension were added to all wells. The final concentrations of chlorhexidine ranged from 0.125 to 64 μg/mL (determined by a preliminary experiment). Well 11 was used as a growth control for bacteria and contained MHB instead of chlorhexidine and 50 μL of the bacterial suspension. Well 12 contained only 50 μL MHB and served as broth control to check the sterility of MHB. This experiment was repeated three times for each isolate and then the microplates were incubated in an incubator with an orbital shaker (Gallenkamp, UK) at 35 °C and 100 rpm for 24 h. The lowest concentration of chlorhexidine that inhibited the visible growth of bacteria was identified as the MIC [2].

### 2.3. Identification of Resistance Genes

The presence of *qac* genes was investigated because they code efflux pumps that are known to be the principal cause of reduced susceptibility to chlorhexidine. The *qacA/B* and *smr* genes are frequently present in enterococci while the *qacE*, *qacEΔ1*, *qacG*, and *IntI 1* genes are widely spread in Enterobacteriaceae. The *qacEΔ1* may also occur in Gram-positive cocci [27,28]. Therefore, enterococci were screened for *qacA/B*, *smr*, and *qacEΔ1* genes and Enterobacteriaceae were screened for *qacE*, *qacEΔ1*, *qacG*, and *IntI 1* genes. The sequences of primer pairs and the methods used to perform PCR to screen the presence of these genes were according to references presented in Table 1. Briefly, bacterial DNA was extracted using the ‘foodproof starprep two kit’ (Biotecon Diagnostics GmbH, Potsdam, Germany) according to the manufacturer’s procedure and the quality and quantity of DNA were checked using a NanoDrop 2000 (Thermoscientific, Waltham, MA, USA). PCR was done by transferring 1 μL of each primer, 22 μL of milliQ water, and 1 μL of the DNA sample to the PCR reaction tubes that contain PCR beads (puReTaq Ready-To-Go PCR beads, GE Healthcare, Amersham, UK). The thermal profiles (Veriti 96-well Thermal cycler, Applied Biosystems, Singapore) for PCR reaction for *qacA/B* and *smr*, *qacG*, *qacE*, and *IntI 1* genes were according to the previous methods (Table 1). Five microliter aliquots of PCR products were analyzed by gel electrophoresis with 2% agarose (Thermo Scientific, TopVision, USA) and 0.5 µg/mL ethidium bromide (Sigma-Aldrich, USA). Gels were visualized by UV using GelDoc (GeneFlash, Syngene, Frederick, Maryland, USA). A 100-bp ladder (Fermetas, O’RangeRuler, Thermo Fisher Scientific, Waltham, MA, USA) was run on each gel as a molecular size marker. The PCR products were sequenced abroad (Macrogen, Seoul, Korea). Alignment and analysis of DNA sequences were done through ChromasPro program (Version 1.41, Technelysium Pty Ltd., South Brisbane, QLD, USA) and then the BLAST program was used to compare sequences online with those found in the NCBI. After submission, the DNA sequences were assigned accession numbers by the European Nucleotide Archive (ENA).

### 2.4. Antibiotic Susceptibility Tests

The antibiotic resistance of Enterobacteriaceae was described in detail in our previous article [26]. All methods for testing and interpretation of the susceptibility of enterococci to antibiotics were from the Clinical and Laboratory Standards Institute (CLSI) [33]. The Kerby Buer disc diffusion method [33] was used to study the susceptibility of enterococci to ampicillin (AMP 10 µg), chloramphenicol (C 30 µg), ciprofloxacin (CIP 5 µg), erythromycin (E 15 µg), penicillin (P 10 µg), and tetracycline (TE 30 µg) (Appendix A). A colorimetric method was used to screen enterococci for β-lactamase production by using nitrocefin discs as described by the manufacturer (Thermo Scientific Remel, Lenexa, KS, USA). These discs are impregnated with nitrocefin which is a chromogenic cephalosporin containing a beta-lactam ring. Beta-lactamase can hydrolyze the amide bond in the beta-lactam ring leading to a change in the color of nitrocefin from yellow to red. To test for β-lactamase, the discs were moistened with sterile water and some bacterial growth (five to six colonies) was smeared on the discs, which were then incubated at room temperature and checked for color change from yellow to red. The reaction was considered negative if there was no change in the color after 60 min.

Vancomycin resistance was investigated using agar dilution with 6 µg/mL vancomycin [33]. In this method, specific concentration of the antimicrobial agent is incorporated into the agar medium while bacterial suspensions are inoculated onto agar surfaces. The appropriate concentration of the antimicrobial agent in the medium is achieved by preparing 10x solution of the required concentration of the antimicrobial agent and then adding one part of this solution to nine parts of the molten agar. A direct colony suspension method was used to prepare bacterial suspension (density of 0.5 McF). A swab was dipped in the suspension, expressed and a quadrant of MHA plate (supplemented with 6 µg/mL vancomycin (Sigma-Aldrich, China)) was streaked. Plates were incubated at 35 °C for 24 h. Growth of >1 colony indicated presumptive vancomycin resistance. Vancomycin resistance was also investigated by determining the MIC of vancomycin using commercial strips (M.I.C. Evaluator VA 256–0.015, Oxoid, Basingstoke, UK) according to the manufacturer’s instructions. The MHA plates were inoculated with bacterial suspensions (density of 0.5 McF) and then vancomycin strips (gradient concentration; 0.015 to 256 µg/mL) were placed on an agar surface and incubated at 35 °C for 24 h. After incubation, the growth inhibition zones were observed and the intersection of the bacterial growth with the strip was identified to obtain the MIC of vancomycin.

Agar dilution was used to search for high-level aminoglycoside resistance against gentamicin and streptomycin [33]. Ten µL of bacterial suspension (density of 0.5 McF) was spotted on Brain Heart Infusion (BHI) agar surface supplemented with either 500 µg/mL gentamicin (gentamicin sulfate salt, Sigma-Aldrich, China) or 2000 µg/mL streptomycin (streptomycin sulfate salt, Sigma-Aldrich, China). The plates were incubated at 35 °C for 24 h for gentamycin or 24–48 h for streptomycin. Growth of >1 colony was considered to indicate resistance to the antibiotic tested. *Enterococcus faecalis* ATCC 51299 was used as a positive control strain and *E. faecalis* ATCC 29212 as a negative control strain for testing resistance to vancomycin, gentamicin, and streptomycin.

### 2.5. Antibiotic Resistance Index (ARI)

The ARI, which can be used to compare multiple antibiotic resistance patterns of the isolated bacteria, was calculated for each bacterial isolate of Enterobacteriaceae and enterococci using this equation:(1)ARI=a/b,
where ‘a’ indicated the number of antibiotics an isolate was resistant to and ‘b’ indicated the total number of antibiotics that were tested against that isolate [34].

### 2.6. Data Analyses

Statistical analysis was accomplished utilizing JMP SAS 12.2.0, USA, and differences were considered significant if *p* was <0.05. A Wilcoxon/Kruskal-Wallis test (nonparametric test applied for distributions that are not normally distributed) was used to test if the source of fresh produce (local or imported) significantly affected the MIC of Enterobacteriaceae and enterococci. The same test was used to analyze if percent resistant enterococci differed significantly according to the type of antibiotic and origin of fresh produce (local or imported). A Spearman’s rank-order correlation was used to determine the relationship between ARI and chlorhexidine MIC for Enterobacteriaceae and enterococci.

## 3. Results

### 3.1. Susceptibility of Enterobacteriaceae and Enterococci to Chlorhexidine

Chlorhexidine MIC for Enterobacteriaceae (their sources can be found in Appendix A) isolated from imported produce (*n* = 50) ranged from 1 to 64 µg/mL and those isolated from local produce (*n* = 38) ranged from 1 to 32 µg/mL (Figure 1). One isolate (2%) of Enterobacteriaceae that was isolated from imported produce (*Raoultella planticola*, source: cabbage, the Netherlands) possessed the highest MIC of 64 µg/mL. Enterobacteriaceae isolated from imported produce also had higher a percentage of bacteria (14%) that produced MIC of 32 µg/mL compared to those isolated from local produce in which 5% of them produced this MIC value (Figure 1). The highest percentage of Enterobacteriaceae bacteria isolated from local produce had MICs of 16 µg/mL (32%) and 2 µg/mL (26%), whereas the highest percentage of Enterobacteriaceae bacteria isolated from imported produce had their MICs at 2 µg/mL (36%) and at the lowest MIC value of 1 µg/mL (18%). Low chlorhexidine MICs of 1 or 2 µg/mL were produced with all *E. coli* isolates including the QC (Quality Control) strain; *E. coli* ATCC 25922. The drug-resistant QC strain *Klebsiella pneumoniae* ATCC BAA-1705 had chlorhexidine MIC of 16 µg/mL. Chlorhexidine MIC for Enterobacteriaceae was not significantly different between bacteria isolated from local and imported produce (Wilcoxon/Kruskal-Wallis, *p* = 0.7087, α = 0.05).

The range of chlorhexidine MIC for enterococci (their sources are shown in Appendix A) isolated from local produce was 1 to 8 µg/mL and for those isolated from imported produce it was 2 to 8 µg/mL (Figure 1). Enterococci that were isolated from local and imported produce had their highest chlorhexidine MICs of 8 µg/mL (8% for local and 11% for imported produce bacteria) and 4 µg/mL (50% for local and 47% for imported produce bacteria) (Figure 1). The highest chlorhexidine MIC for enterococci of 8 µg/mL was found with *E. faecalis* isolates originating from imported lettuce (Jordan and Iran) and from local radish (Appendix A). Chlorhexidine MIC for enterococci was not significantly different between bacteria isolated from local and imported produce (Wilcoxon/Kruskal-Wallis, *p* = 0. 0.8237, α = 0.05). The drug resistant QC strain *E. faecalis* ATCC 51299 had a chlorhexidine MIC of 8 µg/mL, while the drug sensitive QC strain *E. faecalis* ATCC 29212 showed a lesser MIC value of 4 µg/mL (Appendix A).

### 3.2. Resistance Genes in Enterobacteriaceae and Enterococci

The *qacEΔ1* gene was detected in isolates no. 1 and 75, *qacE* in isolates no. 1 and 4, and *qacG* in isolates no. 52, 75 and 84 (Table 2). None of the genes sought in enterococci were detected. The accession numbers of sequences *IntI 1*, *qacEΔ1*, *qacG*, and *qacE* genes are available at http://www.ebi.ac.uk/ena/data/view/LT548573-LT548593 (accessed on 6 September 2022). Sequences of the *IntI 1* genes are also registered in the ‘INTEGRALL’ platform and can be found at http://integrall.bio.ua.pt/?acc=LT548588 and http://integrall.bio.ua.pt/?acc=LT548589 (accessed on 6 September 2022).

### 3.3. Antibiotic Resistance of Enterobacteriaceae and Enterococci

The ARI of Enterobacteriaceae and enterococci are presented in Appendix A. The highest ARI for Enterobacteriaceae was 0.36 and was achieved by nine bacteria (10%) that had resistance to 5 antibiotics. The susceptibility of enterococci to various antibiotics is presented in Appendix A and Figure 2. Higher percentages of enterococci isolated from imported produce were resistant to erythromycin (97%), ciprofloxacin (53%), tetracycline (32%), vancomycin (16%) and chloramphenicol (5%) as compared with those isolated from local produce, which were resistant only to ciprofloxacin (14%) and erythromycin (12%).

All enterococci produced negative results with nitrocefin indicating the absence of β-lactamase. The MIC of vancomycin ranged from 0.25 to 8 µg/mL. No resistance to oxacillin or high-level aminoglycoside (gentamicin and streptomycin) was detected. There was no significant difference in percent resistant enterococci according to the type of antibiotics (Wilcoxon/Kruskal-Wallis, *p* = 0.1617, α = 0.05) or the source (local or imported) of produce (Wilcoxon/Kruskal-Wallis, *p* = 0.1104, α = 0.05). Six isolates (19%) of enterococci were pan-susceptible (susceptible to all antibiotics tested), 19 (61%) were resistant to at least one antibiotic, and 6 (19%) were multidrug resistant (resistant to three antibiotics belonging to different classes) (Figure 3) and had the highest ARI of 0.3 for enterococci (Appendix A).

### 3.4. Association between Bacterial Susceptibility to Chlorhexidine and Their Resistance to Antibiotics

There was a weak positive correlation (Spearman’s rank correlation) between ARI and chlorhexidine MIC which was not statistically significant for either Enterobacteriaceae (*r_s_
*= 0.1504, *n* = 88, *p* = 0.1620) or enterococci (*r_s_
*= 0.3115, *n* = 31, *p* = 0.0880). However, this positive association was statistically significant for enterococci (*r_s_
*= 0.3115, *n* = 31, *p* = 0.0880) when α was considered 0.1 but not for Enterobacteriaceae (*r_s_* = 0.1504, *n* = 88, *p* = 0.1620).

## 4. Discussion

This article is thought to be one of the first that describes the susceptibility of fresh produce-associated bacteria to chlorhexidine. The study showed that all the isolates had chlorhexidine MIC of ≤64 µg/mL. In the food industry, chlorhexidine is used at concentrations of 0.01–0.02% (100–200 µg/mL) and for hand hygiene and personal care, it is used at concentrations of 2–4% (20,000–40,000 µg/mL) [35]. In clinical settings, the chlorhexidine concentration should be at least 10–50 times higher than bacterial in vitro MICs to result in 99.9% death within 10 min at 20 °C [36]. Thus, all isolates in this study had chlorhexidine MIC below the recommended concentration in both food production areas and in clinical settings, but some isolates (those with chlorhexidine MIC ≥ 16 µg/mL) had reduced susceptibility that was not 10 times less than the recommended chlorhexidine MIC in the food industry. It would be important to have disinfectant MIC significantly less than the in-use biocide concentration to ensure their efficient killing. Minimum inhibitory concentration has been used here to measure the susceptibility of planktonic bacterial cells, but these cells may behave differently in real world settings, and this may lead to increased resistance due to factors such as attachment and biofilm production. Indeed, microorganisms were demonstrated to survive after disinfection in food, environmental, and clinical settings [13]. The presence of organic matter, biofilm, and biocide residues at a sublethal concentration may allow for the adaptation and growth of a subpopulation of microbial cells [14,15].

Our finding of chlorhexidine MIC of 1 or 2 µg/mL for all *E. coli* isolates was similar with what was reported for 202 *E. coli* isolates obtained from food animals in Denmark [37]. The agar dilution method conducted in another study [36] yielded the same chlorhexidine MIC of 2 µg/mL for *E. coli* ATCC 25922 as was found in this study. Chlorhexidine MIC for swine *E. coli* ranged from 0.47 to 3.76 µg/mL, and MICs of 0.94 µg/mL or more were regarded as having a reduced susceptibility to chlorhexidine. Clinical *E. coli* were previously found to have a chlorhexidine MIC of 0.5–8 µg/mL [2]. *E. coli* isolated from fresh produce in this study had chlorhexidine MIC that fell within the MIC range of clinical isolates; thus, as a species, they may have reduced susceptibility to chlorhexidine but generally they were more susceptible to chlorhexidine than the other Enterobacteriaceae bacteria tested in this study.

The susceptible chlorhexidine strain *K. pneumoniae* that was used by other investigators [37] yielded a chlorhexidine MIC of 4 µg/mL, while the resistant one produced an MIC of 32 µg/mL. In this study, the drug resistant QC strain *K. pneumoniae* ATCC BAA-1705 gave a chlorhexidine MIC of 16 µg/mL. Chlorhexidine MIC for the 13 *K. pneumoniae* that were isolated in this study ranged from 8–32 µg/mL, with one isolate yielding a chlorhexidine MIC of 32 and most of them (eight isolates) giving an MIC of 16 µg/mL. Thus, in comparison with the other Enterobacteriaceae bacteria, *K. pneumoniae* exhibited reduced susceptibility to chlorhexidine. Some *Enterobacter cloacae* also produced high MIC values of 16 or 32 µg/mL. *E. cloacae* bacteria treated with their MIC chlorhexidine concentration of 7.8 µg/mL were observed by scanning electron microscopy to be bigger in size, having rough surfaces and asymmetric shapes, and many were totally damaged [38].

In this study, the *IntI 1* gene for class 1 integron was detected in 2 *E. coli* isolates; one obtained from local cabbage and the other from imported radish (China), and one *K. pneumoniae* isolated from imported banana (the Philippines). Class 1 integrons are extremely important recruitment platforms (can excise and integrate gene cassettes); in particular, those possessed by clinical bacteria are located on transposons or plasmids which can carry from one to six gene cassettes which in turn carry antibiotic resistance genes or other biocide resistance genes [32,39].

In the current study, *qacE* was detected in 2 *E. coli* isolated from local cabbage and imported lettuce (Jordan). The *qacEΔ1* was detected in 2 isolates; *E. coli* isolated from local cabbage and *Enterobacter ludwigii* from imported cucumber (UAE; United Arab Emirates). The *E. coli* isolate that was obtained from local cabbage harbored *IntI 1*, *qacE*, and *qacEΔ1* and it is possible that the latter two genes are located on an integron. The *qacG* was detected in three isolates; *E. cloacae* isolated from local cabbage, *E. ludwigii* from imported cucumber (the UAE), and *E. cloacae* from imported lettuce (Iran). The low prevalence of *qac* genes found in this study is in concordance with the results of other researchers [40], who looked for the presence of antibiotic and efflux pump genes in Gram-negative bacteria isolated from organic foods. Some studies found that regardless of the presence of *qac* resistance genes in many bacteria, these bacteria were as susceptible to various disinfectants as the isolates that lack them [28]. It was noted that staphylococci may lack efflux-mediated resistance genes but appear phenotypically resistant to chlorhexidine, or they may have resistance genes such as *qacA* but appear susceptible to chlorhexidine. Investigating the RNA expression of chlorhexidine resistance genes during bacterial stress to chlorhexidine can provide better data to elucidate this phenomenon [15]. The discovery of a new gene; *AceI,* that is capable of binding to chlorhexidine and mediating its efflux in *Acinetobacter baumannii* indicates the possibility of the presence of other unknown chlorhexidine resistance genes [16].

The MIC of chlorhexidine for *E. faecium* isolated from fresh produce in Spain was 75 µg/mL for 22 isolates, and the authors considered the isolates to be susceptible as this concentration falls below the recommended level [35]. Like the results of this study, a chlorhexidine MIC range of 0.5–8 µg/mL was achieved with *E. faecalis* and *E. faecium* isolated from food animals, with the majority of the former having it at 8 µg/mL, while the majority of the latter had it at 4 µg/mL [36]. The QC strains *E. faecalis* ATCC 29212 and *E. faecalis* ATCC 51299 had a chlorhexidine MIC of 4 and 8 µg/mL, respectively. Thus, in this study, all enterococci isolates and the QC strains were susceptible to chlorhexidine and had lower MIC values as compared with Enterobacteriaceae bacteria. None of the screened genes of *smr*, *qacEΔ1*, and *qacA/B* were detected in any isolate of enterococci. However, other resistance genes such as *efrAB* were found to be common in enterococci strains isolated from food [40].

Sixty-one percent of the isolated enterococci in this study were resistant to at least one antibiotic, while the proportion of the pan-susceptible and the multi-drug resistant enterococci was equal (19%). In contrast, a higher proportion (34%) of multidrug resistant enterococci was found in fresh produce harvested in the southwestern United States [41]. This could be attributed to the differences in the application of antibiotics in different countries, which might have influenced the prevalence of the multidrug resistant isolates. Moreover, the authors considered resistance to two or more antibiotics as multidrug resistance, while in our study, multidrug resistance was considered for isolates showing resistance against three or more antibiotics belonging to different classes [26]. The results of enterococci resistance to tetracycline were presented in detail in another manuscript [42].

Four isolates (13%) of enterococci showed resistance to erythromycin (a macrolide antibiotic). These included *E. sulfureus* (lettuce, Jordan), *E. casseliflavus* and *E. mundtii* (radish, China), and *E. raffinosus* (local dates). In addition, 17 (55%) enterococci showed intermediate resistance to erythromycin. These included *E. casseliflavus*, *E. faecium*, and *E. faecalis* (imported cabbage, cucumber, lettuce, dates, tomato, and watermelon) and *E. casseliflavus* and *E. faecalis* (local radish, papaya and watermelon). *E. faecium* and *E. faecalis* isolated from fresh produce grown in the USA showed resistance (10 and 3% respectively) and intermediate resistance (75 and 68% respectively) to erythromycin. Erythromycin inhibits protein synthesis by binding to the 50S ribosomal subunit of bacterial cells. Resistance to erythromycin can occur due to different mechanisms such as the presence of *erm* genes that are responsible for methylation of the ribosomal target site or through an efflux mechanism which can be mediated by the *msrA* gene [43]. One isolate (3%) of enterococci; *E. sulfureus* (lettuce, Jordan) showed resistance to chloramphenicol. Fresh produce of the USA was found to have 5% chloramphenicol-resistant *E. faecium* and 3% chloramphenicol-resistant *E. faecalis* [41]. Other researchers did not detect any resistance to chloramphenicol or erythromycin in all enterococci isolated from fresh produce and meats [44].

Seventeen isolates (55%) of enterococci had intermediate resistance to ciprofloxacin. These included *E. faecium*, *E. faecalis* and *E. casseliflavus* that were isolated from local cabbage, radish, mango, papaya, and watermelon, and *E. faecalis* and *E. casseliflavus* that were isolated from imported cabbage, lettuce, radish, dates, and watermelon. Resistance to ciprofloxacin was also shown in 28% of *E. faecium* and 5% of *E. faecalis* isolated from the USA fresh produce while 23% of the former and 21% of the latter showed intermediate resistance to this antibiotic [41]. Three isolates (10%) of enterococci exhibited intermediate resistance to vancomycin in which their MIC was 8 µg/mL. All of them were *E. casseliflavus* and were isolated from cabbage, dates, and watermelon imported from the Netherlands, Saudi Arabia, and Iran, respectively. Likewise, 5% of *E. faecium* isolated from the USA produce showed intermediate resistance to vancomycin [44]. *E. casseliflavus* has been reported to have VanC resistance which is an intrinsic vancomycin resistance that leads to a low-level (MICs 8–16 µg/mL) vancomycin-resistant phenotype [45]. The examined *E. faecalis* ATCC 51299 was resistant to vancomycin and this has been previously shown to be due to the presence of the *vanB* gene [46]. All of the 22 *E. faecium* isolates that were obtained from fresh produce marketed in Spain were susceptible to vancomycin and ampicillin [35].

In this study, all enterococci were susceptible to ampicillin, penicillin, and high levels of gentamicin and streptomycin. In comparison, 3% of *E. faecium* isolated from the USA fresh produce were resistant to streptomycin and 7% were resistant to penicillin. All *E. faecalis* were susceptible to penicillin, gentamycin, and streptomycin and all *E. faecium* were susceptible to gentamycin [41]. Gentamycin and streptomycin resistance of *E. faecalis* ATCC 51299 QC strain was confirmed in this study, and this is known to be due to the presence of *aac(6′)+aph(2″)* and *ant(6)-I* genes, respectively [46]. *E. faecalis* ATCC 51299 also showed resistance to erythromycin and chloramphenicol.

In this study, 68% of enterococci were resistant/intermediate resistant to erythromycin, followed by ciprofloxacin (55%) and then tetracycline (19%). Another study reported the same sequence of frequency of resistant bacteria [24], for *E. faecium* isolated from vegetables. Chloramphenicol, vancomycin, ciprofloxacin, and tetracycline are clinically important drugs for treating enterococcal infections [41]. Thus, detecting various degrees of resistance of bacteria isolated from fresh produce to these antibiotics may raise concerns about the role of the food chain in harboring and spreading such resistant bacteria. The highest ARI of 0.3 was achieved by six enterococci bacteria (13%) that were resistant to three antibiotics. These bacteria included three isolates of *E. casseliflavus*, two isolates of *E. faecalis*, and one isolate of *E. sulfureus* that were isolated from imported cabbage, lettuce, radish, and dates. This may indicate the potential of this group of bacteria to influence the resistome of fresh produce. Although some investigators [24] found a clear separation of enterococcal clinical isolates from those found in the open environments (fresh produce, water and soil) based on the length heterogeneity PCR typing that was caused by the reduced incidence of virulence factors and antibiotic resistance levels in environmental bacteria; however, others [35] suggested that the evolution of hospital adapted pathogens could arise as a result of the selective pressure of hospital environment making them gradually isolated from the environmental ones. In fact, environmental and clinical bacteria were shown to have identical gene cassettes [32].

Results of this study showed that enterococci originated from fresh produce grown locally had low to moderate levels of antibiotic resistance which was not statistically different from antibiotic resistance levels in enterococci isolated from imported produce. This could be due to the geographical spread of antibiotic-resistant bacteria. Finding a low prevalence of antibiotic resistance should not be ignored because it might increase in the future [30]. In fact, multiple antibiotic resistance has increased in Oman [47] in which inappropriate antibiotic prescriptions for patients might have contributed to this increase [48].

There might be an association between the reduced susceptibility to different biocides and the resistance of bacteria to antibiotics. Changes in bacterial cell membranes have been proposed as an important mechanism for a non-specific cross-resistance. However, *qac* genes are often carried on plasmids that can also carry antibiotic-resistance genes. Bacteria that have combined resistance to disinfectants and antibiotics are of major concern in the food industry [6]. Exposure and tolerance to biocides may facilitate the emergence and prevalence of antibiotic-resistant bacteria. A correlation was found between triclosan resistance and resistance to chloramphenicol and nitrofurantoin. Thus, cross-resistance may occur between specific types of biocides and antibiotics [39]. Co-resistance can occur when chlorhexidine resistance genes are located on the same mobile genetic elements as the antibiotic resistance genes. Nurses were found to harbor staphylococci with more resistance genes for chlorhexidine than staphylococci isolated from the general population, which may indicate that the increased uses of chlorhexidine in the hospital environment can select for resistant bacteria [15].

Some researchers [49] found no association between multiple antibiotic resistance in Gram-negative bacteria and resistance to QACs, although *qacE* or *qacEΔ1* were present. Similarly, other investigators [36] found some isolates of *E. faecium* that had high ARI and the lowest biocide MIC. In the present study, Spearman’s rank-order correlation detected a small positive correlation between ARI and MIC for Enterobacteriaceae and enterococci, but it was not statistically significant for both groups. However, this positive association was statistically significant for enterococci when α was considered 0.1 but not for Enterobacteriaceae. Thus, the association between antibiotic resistance and resistance to disinfectants does not appear to be clear yet [15,35,40], and more studies with a larger sample size may better explain it.

## 5. Conclusions

Enterobacteriaceae and enterococci had chlorhexidine MICs below the recommended concentrations in the food production areas and in medicine but some of them had reduced susceptibility that was not 10x less than the recommended levels in the food sector. This might challenge the success of disinfection procedures, where it is necessary to ensure the rapid killing of microbes. Nineteen percent of enterococci were multidrug-resistant and their resistance included some clinically valuable antibiotics. Fresh produce may act as a source or vehicle for spreading antibiotic-resistant bacteria. The presence of *IntI 1*, *qacEΔ1*, *qacE*, and *qacG*, though weakly in Enterobacteriaceae, highlights the necessity of monitoring the antimicrobial resistance trends in produce bacteria in the future.

## Figures and Tables

**Figure 1 foods-11-03085-f001:**
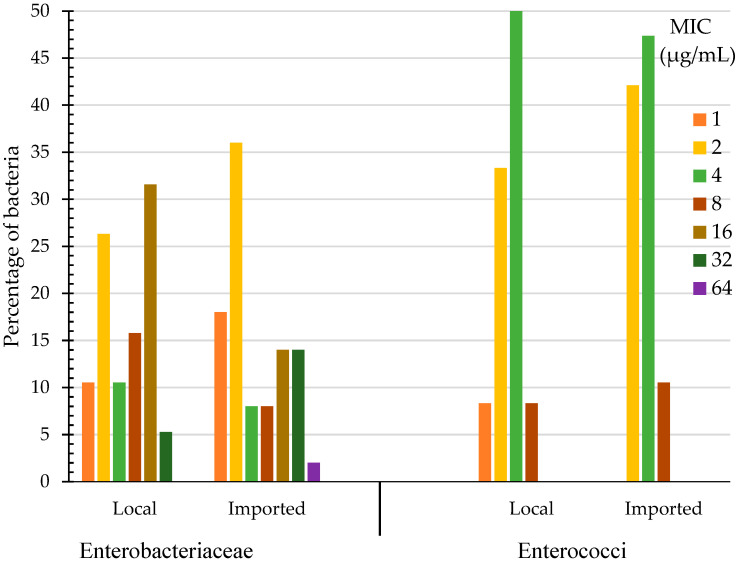
Chlorhexidine Minimum Inhibitory Concentration (MIC) of Enterobacteriaceae and enterococci (%) isolated from local and imported produce.

**Figure 2 foods-11-03085-f002:**
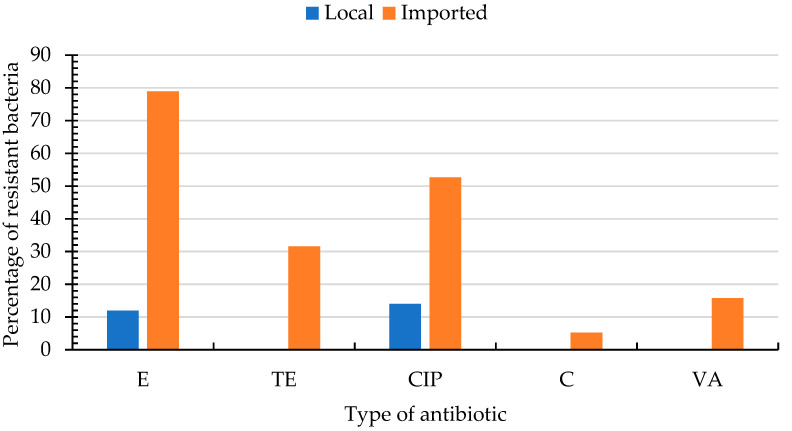
Percentages of resistant enterococci isolated from local (*n*_bacteria_ = 12) and imported (*n*_bacteria_ = 19) fresh produce for different antibiotics. CIP; ciprofloxacin (5 µg), C; chloramphenicol (30 µg), E; erythromycin (15 µg), tetracycline (30 µg), VA; vancomycin (256–0.015 µg). Enterococci were susceptible to AMP; ampicillin (10 µg), GM; gentamicin (500 µg/mL), *p*; penicillin (10 µg), S; streptomycin (2000 µg/mL) and did not hydrolyze nitrocefin (N).

**Figure 3 foods-11-03085-f003:**
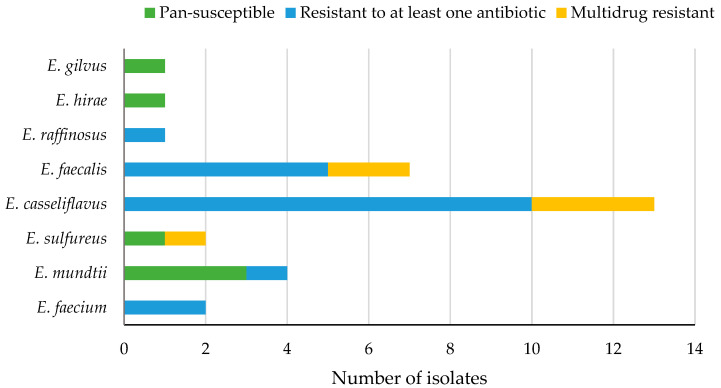
Levels of antibiotic resistance in different species of enterococci (*n* = 31) isolated from fresh produce.

**Table 1 foods-11-03085-t001:** Primers used for the detection of integron integrase class 1 gene (*IntI 1*) and *qac* resistance genes.

Targeted Gene	Primer Sequence 5′–3′	Amplicon Size	Ref.
*IntI 1*	IntA: ATCATCGTCGTAGAGACGTCGG	892	[29,30]
	IntB: GTCAAGGTTCTGGACCAGTTGC		
*qacA/B*	FW: GCAGAAAGTGCAGAGTTCG	360	[31]
	RV: CCAGTCCAATCATGCCTG		
*smr* (*qacC* + *qacD*)	FW: GCCATAAGTACTGAAGTTATTGGA	194	[31]
	RV: GACTACGGTTGTTAAGACTAAACCT		
*qacG*	MRG288: CGCTGATAATGAAGCCGAC	280	[32]
	MRG287: TTGGTTATTTCTGGCTACG		
*qacE*	MRG292: AGCCCCATACCTACAAAG	192	[32]
	MRG291: AGCTTGCCCCTTCCGC		
*qacEΔ1*	FW: GGCTTTACTAAGCTTGCCCC	202	[31]
	RV: AGCCCCATACCTACAAAGCC		

**Table 2 foods-11-03085-t002:** Resistance genes of *IntI 1* and *qac* and their accession numbers as given by the European Nucleotide Archive.

Bacteria No.	Identity (PCR)	Source	Gene	Accession #
1	*E. coli*	Cabbage, Oman	*IntI 1*	LT548588
7	*E. coli*	Radish, China	*IntI 1*	LT548589
94	*K. pneumoniae*	Banana, Philippines	*IntI 1*	-
1	*E. coli*	Cabbage, Oman	*qacE*	LT548593
4	*E. coli*	Lettuce, Jordan	*qacE*	-
1	*E. coli*	Cabbage, Oman	*qacEΔ1*	LT548590
75	*E. ludwigii*	Cucumber, UAE	*qacEΔ1*	-
52	*E. cloacae*	Cabbage, Oman	*qacG*	LT548591
75	*E. ludwigii*	Cucumber, UAE	*qacG*	LT548592
84	*E. cloacae*	Lettuce, Iran	*qacG*	-

- Sequences are not available.

## Data Availability

The data presented in this study are available in this article.

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
