# Peer review of "Antimicrobial Susceptibility of Fresh Produce-Associated Enterobacteriaceae and Enterococci in Oman"

_foods, 2022, doi:10.3390/foods11193085_

Round 1

Reviewer 1 Report

This manuscript reports results from a well-conducted study. The manuscript is well written, and the study design is good. The authors have detected low to moderate levels of antibiotic resistance in enterococci isolates recovered from fresh products. 

The authors do a good job of introducing the background and rationale for their study and the objectives are clearly defined. The materials and methods are clearly and concisely described and appropriate to achieve stated objectives. Moreover, The authors adequately assessed the results with the statistical method.

The results were well presented and discussed objectively and the conclusions reached are supported by the evidence they obtained.

Author Response

No notes are presented as the reviewer kindly accepted the work. Only minor changes were made according to the other reviewer's suggestions.

Reviewer 2 Report

Over all the manuscript is interesting and well written. All the comments are provided in each section and see the highlight texts and the given comment and suggestion in the pdf file.

Author Response

Dear reviewer; 

Please find here the responses to all comments. Please see the attachment (same as shown here). the introduction, results and conclusion were modified according to the suggestions.

Point 1: Consider revising the title to avoid redundancy of words... Enterococci?. I understand what you mean regarding the enterococci. It can be part of the conclusion. Also preferably use antimicrobial susceptibility. All antibiotics are antimicrobials.

 Just  a suggestion: Antimicrobial susceptibility of fresh produce-associated Enterobacteriaceae and Enterococci in xxx country.

Response 1: The title was changed to “Antimicrobial susceptibility of fresh produce-associated Enterobacteriaceae and enterococci in Oman”.

Point 2: add the sources and types of fresh produce at least in summarized form for fruits and vegetables from which the bacteria isolated.

Response 2: The different types of fresh produce were mentioned after the word “bacteria” in the previous sentence to keep the abstract length according to the journal’s requirements as follows: “This study evaluated the susceptibility of fresh produce bacteria (source: banana, cabbage, capsicum, carrots, cucumber, dates, lettuce, mango, papaya, pomegranate, radish, tomato and watermelon) to chlorhexidine and the antibiotic resistance of enterococci”.

Point 3: It is all about susceptibility test and see comments for the title.

Response 3: Antibiotic resistance was changed to susceptibility as follows “Susceptibility of enterococci to various antibiotics was determined using agar dilution, colorimetric, and Kirby-Bauer disc diffusion methods”.

Point 4: Add conclusion about the enterobacterciae it has a practical clinical implication.

Response 4: The following sentence was changed to “Some Enterobacteriaceae bacteria had reduced chlorhexidine MICs that were not 10x less than the recommended concentration (100-200 µg/mL) in food production areas which might challenge the success of the disinfection processes or have clinical impilications if the involved bacteria are pathogenes”.

Point 5: Revise by beginning with brief definition of biocides followed by the antiseptics and disinfectants

Response 5: Done as follows: “Biocides are agents used to kill, inhibit or control the growth of microorganisms”; mentioned in the same reference.

Point 6: Replace with antimicrobials.

Response 6: Done.

Point 7: delete and merge the two citations.

Response 7: Done.

Point 8: delete. This might lead to emergence.

Response 8: Done.

Point 9: if any provide supportive in terms of financial loss from spoilage and morbidity or mortality due to resistance to disinfection.

Response 9: Bacteria that survive the disinfection process can cause food spoilage or affect a food process such as fermentation and thus may have an economic impact [6]. They may also cause disease if they are pathogens and influence the morbidity and mortality rates as they challenge the prevention of disease transmission [7].

One new reference [7] was added (published 2021).

Point 10: Add For instances, before Listeria

Response 10: Done.

Point 11: I do not see the relevance of detailing the mechanism of action here. Delete it.

Response 11: Done. The mechanism of action was deleted.

Point 12: The study?

Response 12: Done; it was changed to “the current study”

Point 13: See  my comments above for the title.

Response 13: Done. It was changed to “The present study aims to determine the susceptibility of fresh produce-associated Enterobacteriaceae and enterococci to various antimicrobials including those that have clinical significance”.

Point 14: Was resistance genes determined only for Enterococci? see the methodology

Response 14: In this paper, yes, the antibiotic resistance was reported only for enterococci but according to comments (points) 1 and 13 to use the term “antimicrobials”. The aim was rewritten as in response 13 to prevent confusion. 

Point 15: Surveillance? rather consider "The findings from the study can...."

Response 15: Done. It was changed to “The findings from this study can”

Point 16: delete and provide justification why the selected genes screened.

Response 16: It was deleted. The justification was written as follows:

“The presence of qac genes was investigated because they code efflux pumps that are known to be the principal cause of reduced susceptibility to chlorhexidine. The qacA/B and smr genes are frequently present in enterococci while qacE, qacEΔ1, qacG, and IntI 1 genes are widely spread in Enterobacteriaceae. The qacEΔ1 may also occur in Gram-positive cocci [26, 27]. Therefore, enterococci were screened for qacA/B, smr, and qacEΔ1 genes and Enterobacteriaceae were screened for qacE, qacEΔ1, qacG, and IntI 1 genes.”

Point 17: Briefly,

Response 17: It was changed to “Briefly”.

Point 18: susceptibility

Response 18: Done. It was changed to susceptibility.

Point 19: Delete it if you did not use another method. Any reason for sing CLSI? why not EUCAST? Was the interpretation based on CLSI?

Response 19: It was deleted. The word interpretation was added to make it clear as follows: “All methods for testing and interpretation of the susceptibility of enterococci to antibiotics were from the Clinical and Laboratory Standards Institute (CLSI).” The inerpretation was based on CLSI and it is reported in Table S1 mentioned in the same paragraph as follows: “The Kerby Buer disc diffusion method [32] was used to study the susceptibility of enterococci to ampicillin (AMP 10 µg), chloramphenicol (C 30 µg), ciprofloxacin (CIP 5 µg), erythromycin (E 15 µg), penicillin (P 10 µg), and tetracycline (TE 30 µg) (Table S1).”

There was no specific reason for using the CLSI except it has been the standard used in our university and its hospital and it is accepted. CLSI and EUCAST are both recommended by the WHO.  I found this quote “The two most commonly used methodologies worldwide are those of the Clinical and Laboratory Standards Institute (CLSI) and the European Committee for Antimicrobial Susceptibility Testing (EUCAST)” by Cusack TP, Ashley EA, Ling CL, Roberts T, Turner P, Wangrangsimakul T, Dance DAB. Time to switch from CLSI to EUCAST? A Southeast Asian perspective. Clin Microbiol Infect. 2019 Jul;25(7):782-785. doi: 10.1016/j.cmi.2019.03.016. Epub 2019 Mar 25. PMID: 30922928; PMCID: PMC6587905.

We will consider using EUCAST in our future research.

Point 20: add the concentration for each antibiotic

Response 20: The concentration of antibiotics is presented between brackets, I added the unit of it for this first mentioning. This is presented in the previous response: “The Kerby Buer disc diffusion method [32] was used to study the susceptibility of enterococci to ampicillin (AMP 10 µg), chloramphenicol (C 30 µg), ciprofloxacin (CIP 5 µg), erythromycin (E 15 µg), penicillin (P 10 µg), and tetracycline (TE 30 µg) (Table S1).” The unit was also added in Figure 2.

Point 21: Data analyses

Response 21: Done. It was changed to data analyses.

Point 22: Merge 3.1 and 3.2 OR  Put  figure 2 at the end of 3.2

Response 22: They were merged.

Point 23: This should not be part of the result. You may add this information in the discussion

Response 23: it was deleted and the sentence rewritten as “The qac EΔ1 gene was detected in isolates no. 1 and 75, qacE in isolates no. 1 and 4 and qacG in isolates no. 52, 75 and 84 (Table 2).

Point 24: Same comment.

Response 24: it was deleted and the following sentence was rewritten as “The ARI of Enterobacteriaceae and enterococci are presented in Table S2.”

Point 25: I do not see the relevance of these two paragraph which are repetition from the introduction. Here you are supposed to reiterate the objectives of the study and mention the major findings that needs discussion in the subsequent paragraphs.  

Response 25: The two paragraphs were deleted. One sentence from conclusion was put as suggested. Another sentence was placed as suggested in point 26.

Point 26: Revise this part by beginning with the key findings followed by an interpretation of the results.... The study showed that all the isolates had chlorhexidine....

Response 26: It was revised as follows: “The study showed that all the isolates had chlorhexidine MIC of ≤ 64 µg/mL.

Point 27: Avoid direct repetition of the results.....rephrase . Our findings for the .....is similar with

Response 27: Our finding of chlorhexidine MIC of 1 or 2 µg/mL for all E. coli isolates was similar with what was reported for 202 E. coli isolates obtained from food animals in Denmark.

Point 28: susceptible.

Response 28: Done. It was changed to susceptible.

Point 29: Unnecessary discussion!!!. It this for the detected genes mentioned below?. As commented above, it is advisable to begin with the interpretation of your finding and followed by comparison with literature and then possible explanation for the convergence and divergence of your findings.

Response 29: done. The unnecessary paragraph was deleted.

Point 30: Repetition of the results! Starting with the next sentence is enough.

Response 30: The repetition was deleted.

Point 31: Again repetition of results!!

Response 31: It was deleted.

Point 32: Either delete or present the results for the difference between gram positive and gram negative in the result section.

Response 32: it was deleted.

Point 33: Too general! here you are in the discussion section! Deleting is suggested.

Response 33: it was deleted.

Point 34: This needs further discussion. You are supposed to compare your finding with others and explain the possible reason for the difference and agreement.

Response 34: Further comparison and explanation was added as follows

In contrast, higher proportion (34%) of multidrug resistant enterococci was found in fresh produce harvested in the southwestern United States (44). This could be attributed to the differences in the application of antibiotics in different countries which might have influenced the prevalence of the multidrug resistant isolates. Moreover, the authors considered resistance to two or more antibiotics as multidrug resistance while in our study, multidrug resistance was considered for isolates showing resistance against three or more antibiotics belonging to different classes (5).

Point 35: see previous comment

Response 35: It was deleted.

Point 36: Discussion Or Conclusion??

Response 36: part of it was deleted and paraphrased as conclusion as follows: “Fresh produce may act as a source or vehicle for spreading antibiotic-resistant bacteria.”

Point 37: Delete from here and may consider putting at the beginning of the discussion

Response 37: It was deleted from here and put at the beginning of the discussion.
